# The pulmonary mycobiome—A study of subjects with and without chronic obstructive pulmonary disease

**Einar M. H. Martinsen**[1]*, **Tomas M. L. Eagan**[1,2], **Elise O. Leiten**[1], **Ingvild Haaland**[1], **Gunnar R. Husebø**[1,2], **Kristel S. Knudsen**[2], **Christine Drengenes**[1,2], **Walter Sanseverino**[3], **Andreu Paytuví-Gallart**[3], **Rune Nielsen**[1,2]

1 Department of Clinical Science, University of Bergen, Bergen, Norway, 2 Department of Thoracic Medicine, Haukeland University Hospital, Bergen, Norway, 3 Sequentia Biotech SL, Barcelona, Spain

* einar.martinsen@uib.no

## Abstract

### Background

The fungal part of the pulmonary microbiome (mycobiome) is understudied. We report the composition of the oral and pulmonary mycobiome in participants with COPD compared to controls in a large-scale single-centre bronchoscopy study (MicroCOPD).

### Methods

Oral wash and bronchoalveolar lavage (BAL) was collected from 93 participants with COPD and 100 controls. Fungal DNA was extracted before sequencing of the internal transcribed spacer 1 (ITS1) region of the fungal ribosomal RNA gene cluster. Taxonomic barplots were generated, and we compared taxonomic composition, Shannon index, and beta diversity between study groups, and by use of inhaled steroids.

### Results

The oral and pulmonary mycobiomes from controls and participants with COPD were dominated by *Candida*, and there were more *Candida* in oral samples compared to BAL for both study groups. *Malassezia* and *Sarocladium* were also frequently found in pulmonary samples. No consistent differences were found between study groups in terms of differential abundance/distribution. Alpha and beta diversity did not differ between study groups in pulmonary samples, but beta diversity varied with sample type. The mycobiomes did not seem to be affected by use of inhaled steroids.

### Conclusion

Oral and pulmonary samples differed in taxonomic composition and diversity, possibly indicating the existence of a pulmonary mycobiome.

**Data Availability Statement:** The dataset and code supporting the conclusions of this article is available in the DRYAD repository. Age and sex are omitted from the metadata due to privacy

concerns. Available from: https://doi.org/10.5061/dryad.w3r2280nz.

**Funding:** The MicroCOPD study was funded by unrestricted grants and fellowships from Helse Vest, GlaxoSmithKline, Bergen Medical Research Foundation, and the Endowment of Timber Merchant A. Delphin and Wife through the Norwegian Medical Association. The MicroCOPD funders had no role in study design, data collection and analysis, decision to publish, or preparation of the manuscript. Sequentia Biotech SL provided support in the form of salaries for authors Walter Sanseverino and Andreu Paytuví-Gallart, but not study design or data collection. All funding of data collection and laboratory analyses were from the MicroCOPD Study. Walter Sanseverino and Andreu Paytuví-Gallart both contributed to interpretation of results, and preparation of the manuscript. The specific roles of all authors are articulated in the 'author contributions' section.

**Competing interests:** I have read the journal's policy and the authors of this manuscript have the following competing interests: Einar M. H. Martinsen, Elise O. Leiten, Ingvild Haaland, Gunnar R. Husebø, Kristel S. Knudsen, and Christine Drengenes declare no conflict of interest. Walter Sanseverino and Andreu Paytuví-Gallart are employed at Sequentia Biotech SL. Rune Nielsen reports grants from the Endowment of Timber Merchant A. Delphin and Wife (The Norwegian Medical Association) and grants from GlaxoSmithKline during the conduct of the study, and grants and personal fees from AstraZeneca, grants and personal fees from GlaxoSmithKline, grants and personal fees from Boehringer Ingelheim, and grants from Novartis outside the submitted work. Tomas M. L. Eagan reports grants from Helse Vest (Western Norway Regional Health Authority) during the conduct of the study, and personal fees from Boehringer Ingelheim outside the submitted work. This does not alter our adherence to PLOS ONE policies on sharing data and materials.

## Introduction

Fungi are ubiquitous, and are found in indoor and outdoor environments [1]. Due to its direct communication with surrounding air, the respiratory tract is constantly exposed to fungal spores through inhalation [2]. Healthy airways possess effective removal of such spores through mucociliary clearance and phagocytosis. In contrast, impaired defence mechanisms, use of immunosuppressant, and frequent use of antibiotics probably predispose for increased fungal growth [2], and all factors are quite frequent in chronic obstructive pulmonary disease (COPD).

The fungal part of the microbiome, the mycobiome, of the lungs is understudied [3], and only three studies have used next generation sequencing to study the mycobiome of the respiratory tract in COPD particularly [4–6]. Notably, participants in Cui et al.´s study were also HIV infected, and only ten had COPD [4]. The study by Su et al. [5] and Tiew et al. [6] used sputum samples, which are vulnerable to contamination from the high-biomass oral cavity. By contrast, mycobiome studies of other respiratory diseases have evolved rapidly. For instance, a study on asthma patients showed higher fungal burdens in participants receiving corticosteroid therapy [7], while another study has revealed associations between *Aspergillus*-specific immunisation and bronchiectasis severity [8]. There is clearly a need for large studies of the mycobiome, with a well-characterised COPD disease population and healthy controls.

The Bergen COPD Microbiome study (short name "MicroCOPD") fills this scientific void [9]. Samples were collected from the lower airways of participants with and without COPD using bronchoscopy. The aim of the current paper was threefold: 1) To characterise and compare the oral and pulmonary mycobiomes in a large cohort of participants without lung disease (controls). 2) To characterise the oral and pulmonary mycobiomes of participants with COPD, and contrast it to controls, and finally, 3) To examine whether the mycobiomes were affected by the use of inhaled steroids (ICS) in participants with COPD.

## Materials and methods

### Study design and population

The study design of the MicroCOPD study has previously been published [9]. MicroCOPD was a single-centre observational study carried out in Bergen, Western Norway. Study enrolment was between April 11th, 2013, and June 5th, 2015. The study was conducted in accordance with the declaration of Helsinki and guidelines for good clinical practice. The regional committee of medical ethics Norway north division (REK-NORD) approved the project (project number 2011/1307), and all participants provided written consent.

Both subjects with and without COPD were invited to participate. Participants from two previous cohort studies in our vicinity, the GeneCOPD study and the Bergen COPD cohort study, were contacted regarding participation in the current study, and some participants were recruited through media, the local outpatient clinic, or among hospital staff [9]. Potential participants were excluded if they had increased bleeding risk, unstable cardiac conditions, hypercapnia, or hypoxaemia when receiving oxygen supplement, as specified in the study protocol [9]. We postponed participation for subjects that had used antibiotics or systemic steroids last 2 weeks prior to participation, and COPD patients should not have been admitted to hospital due to COPD last 2 weeks. Furthermore, participants with symptoms of an ongoing systemic or respiratory infection could not attend, but had to postpone participation. COPD was defined as chronic airway obstruction (low $FEV_1/FVC$) in presence of respiratory symptoms [10], and the diagnosis was verified by experienced pulmonologists based on spirometry, radiologic imaging, respiratory symptoms, and disease history. Subjects without COPD or

other lung diseases were defined as control subjects. 22 control subjects had a ratio of $FEV_1$/FVC lower than 0.7, but did not have symptoms of COPD.

## Data collection

All data collection was performed in our outpatient clinic. A post-bronchodilator spirometry was performed before the bronchoscopy. Study personnel conducted a structured interview regarding contraindications, medication use, comorbidities, smoking habits, and evaluation of dyspnoea. A sterile unsealed bottle of phosphate-buffered saline (PBS) was opened prior to the procedure, and the fluid within was used for all sample fluids, including negative control samples, oral wash (OW), and bronchoalveolar lavage (BAL). The OW sample was taken before the bronchoscopy by gargling 10 ml of the PBS water for 1 minute; collected in a sterile Eppendorf tube. The bronchoscopy was performed with the participant in supine position using oral access. Topical anaesthesia was given by a 10 mg/dose lidocaine oral spray pre-procedurally, and 20 mg/ml lidocaine was delivered per-operatively through a catheter within the bronchoscope's working channel. Light sedation with alfentanil was offered to all. The details of bronchoscopic sampling have been published previously [9]. The yields of two fractions of protected BAL of 50 mL were collected from the right middle lobe using a sterile catheter (Plastimed Combicath, prod number 58229.19) inserted in the bronchoscope working channel. The second fraction was used for the current mycobiome analysis. Additionally, a sample from the PBS was taken for each participant directly from the bottle used for that particular participant, without entering the bronchoscope or participant. This PBS sample served as a negative control sample.

## Laboratory processing

Fungal DNA was extracted using a combination of enzymatic lysis with lysozyme, mutanolysin, and lysostaphin, and mechanical lysis methods using the FastPrep-24 as described by the manufacturers of the FastDNA Spin Kit (MP Biomedicals, LLC, Solon, OH, USA). Libraries were prepared with a modified version of the Illumina 16S Metagenomic Sequencing Library Preparation guide (Part no. 15044223 Rev. B). The internal transcribed spacer (ITS) 1 region was PCR amplified (increased from 25 to 28 cycles) using primer set ITS1-30F/ITS1-217R, which sequences are GTCCCTGCCCTTTGTACACA and TTTCGCTGCGTTCTTCATCG [11]. A subsequent index PCR was performed with 9 cycles instead of 8. Samples underwent 2x250 cycles of paired-end sequencing in three separate sequencing runs on Illumina HiSeq (Illumina Inc., San Diego, CA, USA).

## Bioinformatics

Quantitative Insights into Microbial Ecology (QIIME) 2 [12] version 2019.01 and 2019.10 was chosen as the main pipeline for bioinformatic analyses, and additional R packages were utilised as suited [13]. FASTQ files containing all fungal reads were trimmed using the q2-itsxpress plugin [14]. Trimmed reads were then denoised, i.e., identification and removal of low-quality reads and chimeric sequences, using the Divisive Amplicon Denoising Algorithm version 2 (DADA2) q2-dada2 plugin [15]. DADA2 also generated exact amplicon sequence variants (ASVs). LULU, an R package to curate DNA amplicon data post clustering, was used to exclude artefactual ASVs [16]. ASVs present in only one sample, and ASVs with a total abundance less than 10 sequences across all samples, were filtered out. Presumed contaminants were identified using the R package Decontam [17] with the prevalence-based approach (user defined threshold = 0.5), and then removed. Taxonomic assignments were made using a UNITE database for fungi with clustering at 99% threshold level [18] (via q2-feature-classifier

[19] classify-sklearn [20]). Resulting ASVs assigned only as *Fungi* at kingdom level, or *Fungi* at kingdom level with unidentified phylum, were manually investigated using the BLASTN program in NCBI [21]. ASVs with unambiguous BLASTN results with a high max score were repeatedly assigned to new taxonomic assignments using UNITE databases with fungi or all eukaryotes with different threshold levels [18, 22–24] (via q2-feature-classifier [19] classify-sklearn [20] and classify-consensus-blast [25]), and included for further analyses if the new assignments matched the BLASTN result. ASVs with ambiguous or non-fungal BLASTN results were discarded. Alpha diversity was calculated using Shannon index, and beta diversity metrics (Bray-Curtis dissimilarity and Jaccard similarity coefficient) were estimated using q2-diversity after samples were rarefied (subsampled without replacement) to 1000 sequences per sample. The rarefaction depth was chosen based on testing with multiple different values and resulting alpha rarefaction plots. We aimed to find a rarefaction depth as high as possible while excluding a minimum of samples.

## Data analyses

Statistical analyses of demographical data were analysed with Stata version 15 [26]. A flow chart of the bioinformatic process was generated using Flowchart Designer version 3 (http://flowchart.lofter.com). Alpha- and beta diversity analyses including participants with COPD were stratified by GOLD stage [10]. Statistical differences in alpha diversity measured with Shannon index were tested with R using Kruskal-Wallis for unpaired variables, and Wilcoxon signed-rank test for paired analyses. Differences in beta diversity between study groups and ICS use were tested with permuted analysis of variance (PERMANOVA) adjusted for sex, age, and percentage of predicted forced expiratory volume in 1 second ($FEV_1$), and differences in spread with permuted multivariate analysis of beta-dispersion (PERMDISP). Procrustes analyses were performed to check for differences in beta diversity between sample types. PERMANOVA, PERMDISP, and Procrustes were analysed using the Vegan package in R [27]. We used both Bray-Curtis and Jaccard distances for the beta diversity analyses. To compare taxonomic composition between pairs of samples we calculated the Yue-Clayton measure of dissimilarity (1-θYC) [28]. Furthermore, differences in distributions and relative abundances were evaluated by the Microbiome Differential Distribution Analysis (MicrobiomeDDA) omnibus test [29], the second version of analysis of composition of microbiomes (ANCOM v2, https://github.com/FrederickHuangLin/ANCOM) [30], and the second version of ANOVA-Like Differential Expression (ALDEx2) [31–33] at genus level. A significance level of 0.05 was used in all analyses.

## Results

### Demographics of participants

The majority of participants with COPD was regular ICS users, and presented with more comorbidities, higher medication use, and poorer lung function measurements ($FEV_1$/FVC-ratio and mMRC) than controls (Table 1).

**Flow chart.** The bioinformatic processing is shown in Fig 1, and details are given in S1 File. ASVs identified by Decontam as presumed contaminants are listed in S1 Table.

**Taxonomy and abundance/distribution testing.** The taxonomic composition of the OW and BAL mycobiomes are displayed on group level in the rank abundance plots (Fig 2A and 2B). Both controls and participants with COPD were dominated by *Candida*, particularly in the OW samples, reaching nearly 80% of total mean relative abundance. The relative abundances of *Malassezia* and *Sarocladium* were high in the control and COPD groups. We also plotted percentage of reads belonging to either *Basidiomycota* or *Ascomycota* (S1 Fig). There

**Table 1. Demographics of participants providing fungal samples in the MicroCOPD study.**

| Variable | Control, n = 100 | COPD, n = 93 |
|---|---|---|
| Age, mean years (SD) | 65.6 (8.5) | 67.5 (7.6) |
| Male, sex (%) | 57 (57.0) | 50 (53.8) |
| Number of medications, mean (SD) | 1.8 (1.7) | 5.2 (3.1) |
| Use of inhaled steroids (%) | - | 56 (60.2) |
| Number of comorbidities, mean (SD) | 0.8 (1.0) | 1.4 (1.2) |
| $FEV_1$, mean % of predicted (SD) | 104.0 (12.3) | 61.1 (17.3) |
| FVC, mean % of predicted (SD) | 111.7 (13.6) | 98.6 (18.7) |
| $FEV_1$/FVC-ratio, mean (SD) | 0.7 (0.1) | 0.5 (0.1) |
| Pack years, mean (SD) | 16.6 (14.3) | 30.0 (17.9) |
| Smoking status (%) | | |
| Daily | 24 (24.0) | 22 (23.7) |
| Ex-smokers | 58 (58.0) | 70 (75.3) |
| Never | 18 (18.0) | 1 (1.1) |
| mMRC Grade 2 and higher (%)* | | |
| Grade 2: Dyspnoea when walking at level ground | 3 (3.0) | 16 (17.6) |
| Grade 3: Dyspnoea when walking 100 meters | - | 12 (13.2) |
| Grade 4: Dyspnoea at rest | - | 2 (2.2) |

$FEV_1$: forced expiratory volume in 1 second, FVC: forced vital capacity, mMRC: modified medical research council dyspnoea scale.

* Two participants with COPD missed information on mMRC.

seemed to be a tendency towards higher proportions of *Basidiomycota* in the COPD group compared to the control group both in the OW and BAL plot. No consistent differences were found between study groups in terms of differential abundance/distribution, and results varied between available statistical tests (S2 Table).

Taxonomy is displayed for each individual participant in Fig 3A and 3B, enabling us to compare OW and BAL samples from this particular participant. We observed intra-individual differences between the sample types for the plotted taxonomic levels (phylum and genus). This was elaborated with Yue-Clayton testing for each OW/BAL/negative control sample pair (S2 Fig). The Yue-Clayton measure is 0 with perfect similarity and 1 with perfect dissimilarity. The average Yue-Clayton measure from OW and BAL samples was 0.63, and 121 out of 180 sample pairs had a Yue-Clayton measure above 0.2. ANCOM and ALDEx2 (S2 Table) found *Candida* to differ in abundance between OW and BAL, also stratified by smoking status and ICS usage. The taxonomy of regular ICS users and non-ICS users was not easily distinguishable (Fig 3A and 3B), and no consistent differences were seen in differential abundance/distribution testing (S3 Table).

## Diversity

We found no significant differences in alpha diversities between the different study groups or ICS usage in BAL samples, nor between BAL and OW samples (Fig 4A and 4B, S3 Table). Beta-diversity results resembled those of alpha diversity (S3 Fig). However, principal coordinates analysis (PCoA) plots before and after symmetric Procrustes transformation (S4 Fig), indicated that there were significant differences in the composition between OW and BAL samples from the same individual. The Procrustes transformation yields a sum of squared distances (M^2) that specifies how similar sample pairs are. Generally, a M^2 above 0.3 is

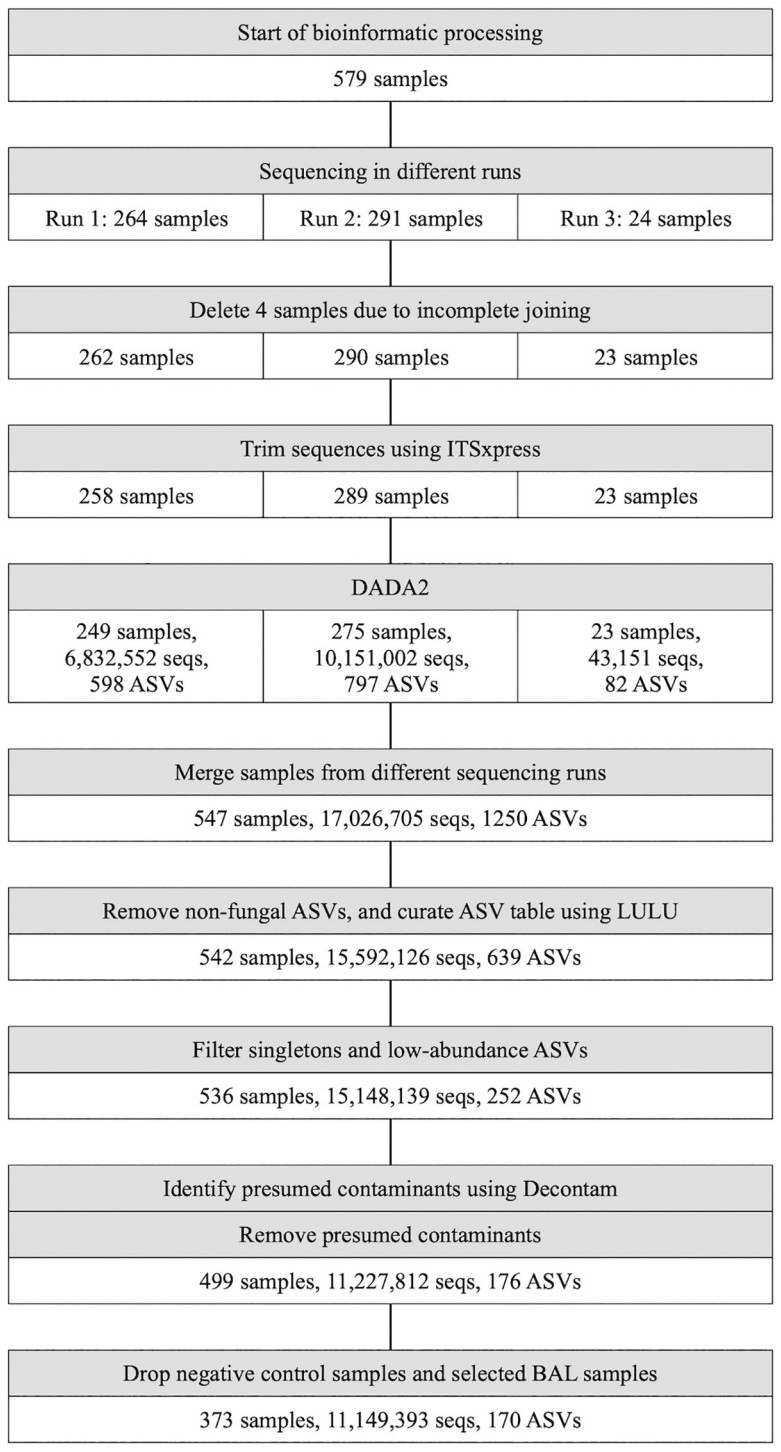

**Fig 1. Flow chart of fungal samples, sequences, and fungal ASVs in the MicroCOPD study.** DADA2: Divisive Amplicon Denoising Algorithm version 2, seqs: sequences, ASV: amplicon sequence variant. Samples were sequenced in three different runs before trimming and denoising. Data from different sequencing runs were merged, and then further processed to exclude presumed contaminants and negative control samples prior to analyses.

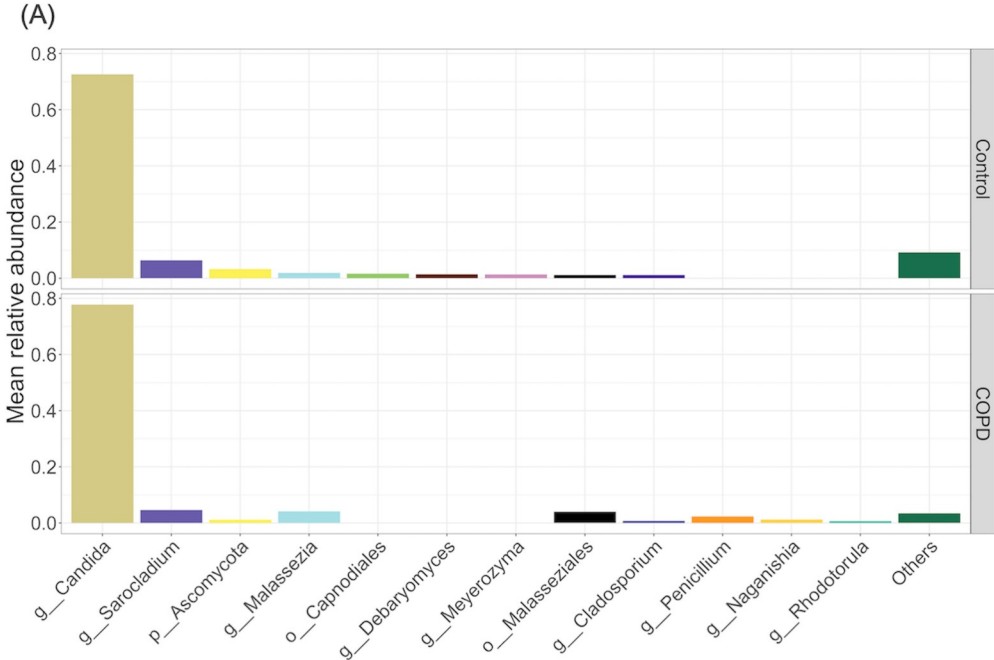

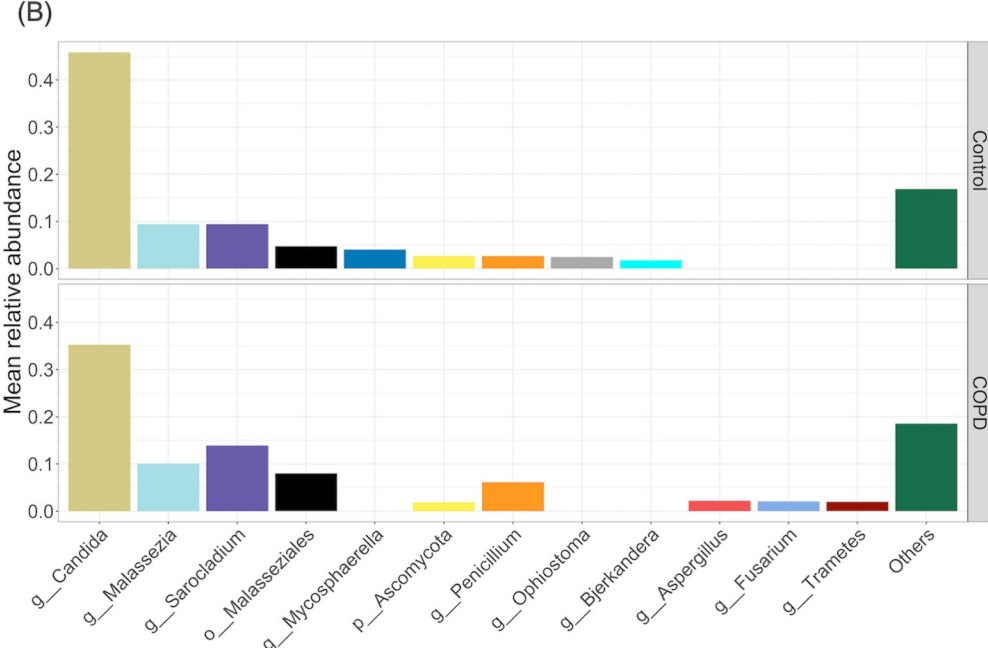

**Fig 2.** Rank abundance plots using most abundant fungi in (A) oral wash and (B) bronchoalveolar lavage. Plots display the nine most abundant taxa in each group. Remaining, low abundance taxa are merged in the "Others" category. Not all sequences could be assigned taxonomy at the genus level and are therefore displayed as o__*Malasseziales*, p__*Ascomycota*, and o__*Capnodiales*.

interpreted as unsimilar. OW and BAL samples clustered differently, and M^2 were 0.953 and 0.8958 for Bray-Curtis and Jaccard, respectively. However, this was statistically significant only for Jaccard (p = 0.003).

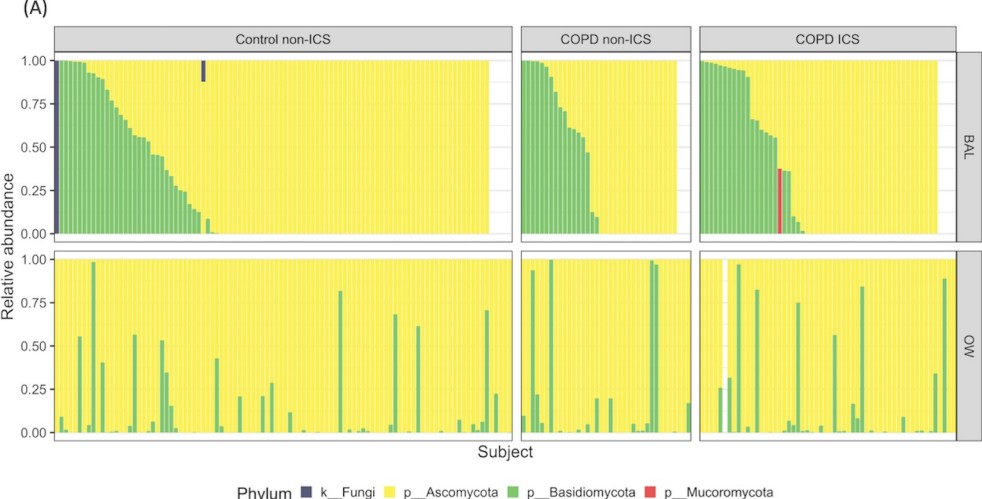

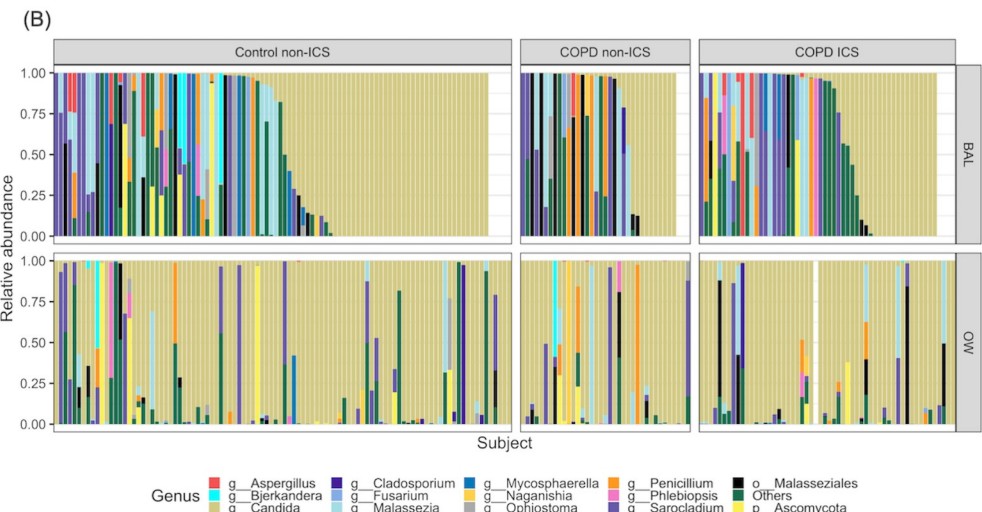

**Fig 3.** Most abundant fungal taxonomic assignments at (A) phylum level and (B) genus level. ICS: inhaled steroids, BAL: bronchoalveolar lavage, OW: oral wash. Taxa are sorted on *Ascomycota* in bronchoalveolar lavage samples in Fig 3A, and *Candida* in bronchoalveolar lavage in Fig 3B. Each column represents a sample, and columns from BAL and OW corresponds to each other. That means, a BAL column and the corresponding OW column below show samples from the same participant. Not all sequences could be assigned taxonomy at the phylum or genus level and are therefore displayed as *k__Fungi*, *p__Ascomycota* or *o__Malasseziales*.

## Discussion

We have reported the oral and pulmonary mycobiome in a large bronchoscopy study, the first of its kind on non-immunocompromised patients and with a large healthy control population. The mycobiomes were dominated by *Candida*, and there were more *Candida* in OW compared to BAL for both study groups. Observed differences in taxonomic composition were not consistent between three different differential abundance/distribution tests. There was no difference in diversity between study groups. No apparent effects were seen on the mycobiomes from ICS usage.

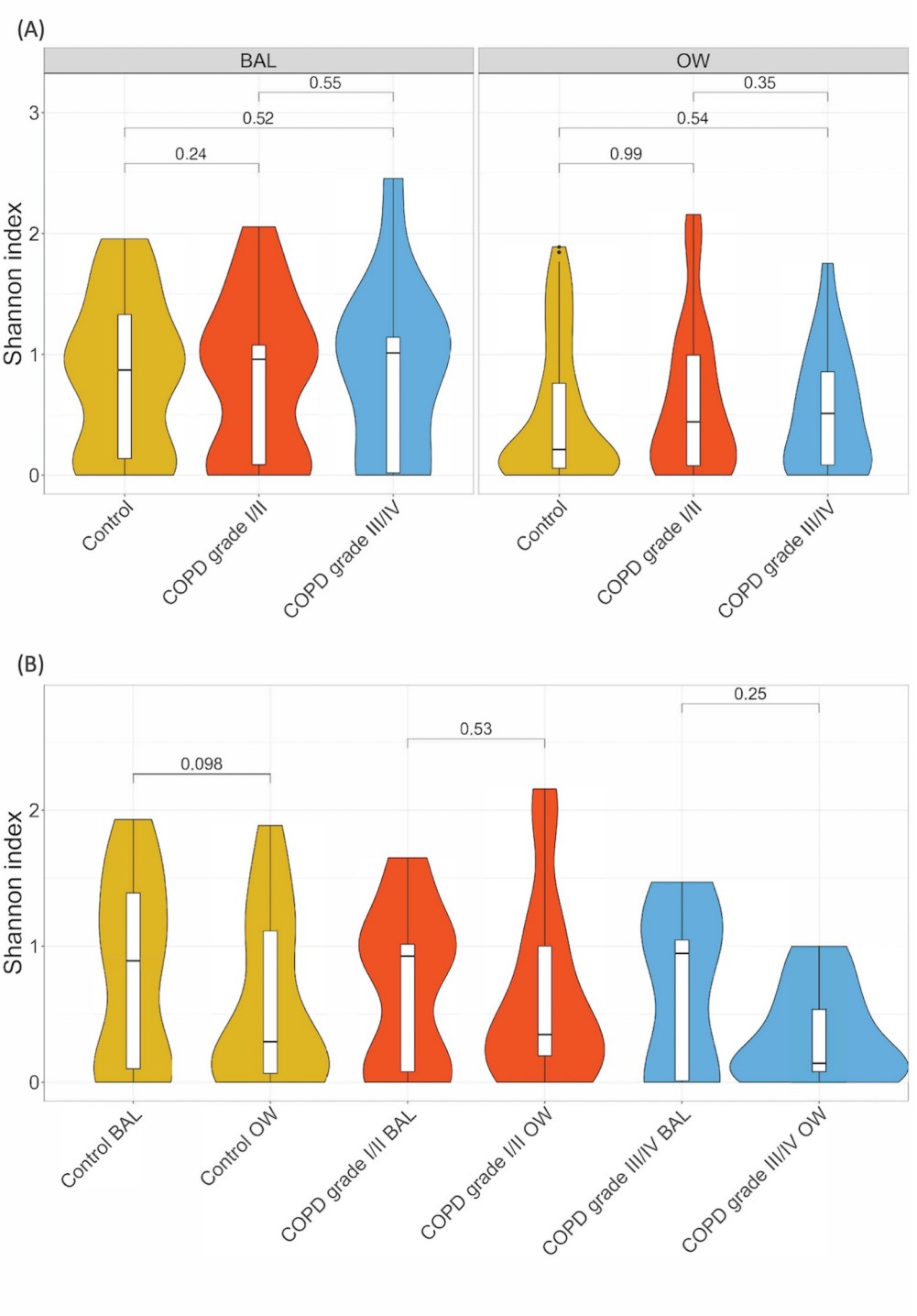

**Fig 4.** Alpha diversity plots and comparisons between (A) study groups and (B) sample types. BAL: bronchoalveolar lavage, OW: oral wash. Alpha diversity was evaluated using Shannon index. Statistical differences in alpha diversity were tested using Kruskal-Wallis for unpaired variables (between study groups), and Wilcoxon signed-rank test for paired analyses (between sample types). Number of samples in each group was as follows: Fig 4A (unpaired): Control BAL: 40, control OW: 76, COPD grade I/II BAL: 29, COPD grade I/II OW: 50, COPD grade III/IV OW: 23, COPD grade III/IV BAL: 12. Fig 4B (paired): Control BAL: 31, control OW: 31, COPD grade I/II BAL: 24, COPD grade I/II OW: 24, COPD grade III/IV OW: 9, COPD grade III/IV BAL: 9.

We observed a high *Candida* load in OW samples from controls, in good agreement with previously published studies [34–36]. None of these studies included more than 20 individuals. Thus, our data from 100 controls adds valuable data to this field. The healthy lung mycobiome is reported to be highly variable between individuals [4, 6, 7, 37–39]. Some of the most abundant taxa are *Candida* [6], *Davidiellaceae* [37], *Cladosporium* [37–39], *Saccharomyces* [4, 6], *Penicillium* [4], *Debaryomyces* [38], *Aspergillus* [7, 37], *Eremothecium* [39], *Systenostrema* [39], and *Malasseziales* [7]. The listed studies include few participants, ranging from 10 [7] to 47 [6]. BAL samples from controls in the current study were dominated by *Candida*, followed by *Malassezia* and *Sarocladium*. That *Candida*, one of the most well-known fungal pathogens [40], resides in the lungs of healthy individuals is clinically interesting. It has been shown that colonising *Candida* in the gut could become invasive due to certain triggers [41], and similar mechanisms are not unlikely to happen in the lungs. Primer bias might explain some of the observed differences between our study and previous studies [42], and our chosen primer set has shown improved coverage of *Candida* compared to the ITS1 –ITS2 primer set used by two [37, 38] of the listed papers above [11]. Furthermore, *Malassezia* are common skin commensals, and despite protected sampling and the bioinformatic contamination removal (Decontam), we cannot exclude contamination *per se*. Also, some reports indicate that some extraction protocols and primers might be less suited to *Malassezia* [36, 43]. Different DNA extraction methods and primers could thus explain the observed differences in *Malassezia* proportions between our study and others.

OW samples differed from the BAL samples for all measures including taxonomy, Yue-Clayton measures, and beta diversity. Cui et al. reported that OW and BAL overlapped in PCoA plots from healthy individuals, while induced sputum (IS) samples clustered separately [4]. In agreement with our result, they also found more *Candida* in OW and IS, compared to BAL. They discovered that 39 fungal species were disproportionately more abundant in the BAL and 203 species in the IS, as compared with the OW. We could not replicate this latter finding, but differences could be explained by the different methodologies applied.

Only three previous studies have explored the lung mycobiome in COPD [4–6]. Cui et al. found that the lung mycobiome in HIV-infected individuals with COPD (n = 10) was associated with an increased prevalence of *Pneumocystis jirovecii*, as compared to HIV-positive individuals without COPD (n = 22) [4]. No *Pneumocystis* was observed in our data. However, *Pneumocystis* is known to be associated with HIV, and the *Pneumocystis* genome only includes one copy of the ITS1 locus, which could result in a negative sequencing result [44].

Both Su et al. [5] and Tiew et al. [6] collected sputum samples from COPD patients. When Tiew compared to healthy subjects they found high abundances of *Candida* in both groups, but also found increased alpha diversity in COPD [6]. Su investigated samples during exacerbations, and found *Candida*, *Phialosimplex*, *Aspergillus*, *Penicillium*, *Cladosporium*, and *Eutypella* [5]. Both studies utilised sputum samples, which hampers direct comparison to our BAL samples. Indeed, IS samples have been shown to cluster separately from BAL samples in PCoA ordinations [4].

Few differences were seen when we compared the mycobiomes from controls and participants with COPD. However, hypothesis testing of microbiome compositional data is an ongoing research area without standardisation. Thus, we chose to perform three different tests with different foundations. ANCOM v2 and ALDEx2 agreed there was no significantly differential abundant taxa between study groups. MicrobiomeDDA tests the difference in the entire distribution, taking abundance, prevalence, and dispersion all into account, and detected significant taxa differences between study groups both in OW and BAL (S2 Table). These conflicting results complicate a general conclusion.

Some studies on inflammatory bowel disease, and cystic fibrosis (CF) have found dysbiosis to be expressed in terms of the *Basidiomycota* to *Ascomycota* ratio [45, 46]. Most known fungal pathogens are found in the *Ascomycota* phylum. In our study, medians of the *Basidiomycota* to *Ascomycota* ratios were all 0 from different study groups in OW and BAL separately, in line with the 0.03 median found in a CF study [46]. That means, despite a higher *Basidiomycota* fraction in COPD compared to controls in our data (S1 Fig), the majority of samples were dominated by *Ascomycota*.

Some limitations of the current study deserve mentioning. First, a longitudinal study with analyses on interactions between fungi and other kingdoms, and between fungi and host responses, could have provided more insight into the details of the COPD mycobiome. Secondly, contamination is particularly problematic for mycobiome studies because of airborne particles, and samples from the lower respiratory tract are especially vulnerable due to the low biomass. We countered this by using protected sampling methods, and collecting negative control samples, subject to the same laboratory protocol as the procedural samples, for each procedure. These samples were used for contamination removal through the R package Decontam, and subject to detailed analyses (S2 Fig, S4 Table and S5 Fig). Third, we did not include positive controls or mock communities in our project. Fourth, ITS primers are biased [42, 43], possibly explaining the low prevalence of *Aspergillus* and difficulties identifying *Yarrowia lypolytica* in our data. Still, ITS is the recommended marker-gene region for fungal studies [47], though no consensus seems to prevail whether ITS1 or ITS2 should be used [43, 48, 49]. Fifth, all mycobiome studies suffer from a fungal dual naming system [50], and also suffer from incomplete reference libraries and inconsistencies due to taxonomic reassignments [2]. We manually reviewed every sequence assigned only as "*k__Fungi*" to secure the best possible taxonomy. Finally, confounding factors and batch effects could not be ruled out. We included a thorough examination of diversity and differential abundance/distribution testing to look for potential confounding effects from several important clinical parameters (S3 Table). No apparent effects were seen. We observed no statistically significant difference in alpha diversity between sequencing runs (S5 Fig), but there might have been an effect on beta diversity (S6 Fig), particularly between sequencing run 1 and 2 (S5 Table). In terms of differential abundance/distribution, it seemed that *Sarocladium* differed in abundance/distribution between sequencing run 1 and 2 (S7 Table).

Studies on the lung mycobiome are still in their infancy, and results from the current study add knowledge to an understudied area. Samples from the mouth differed from pulmonary samples both in controls and participants with COPD, which may indicate the existence of a pulmonary mycobiome. Certain inferences on taxonomic compositions differences between study groups could not be made due to inconsistent results among the differential abundance/distribution tests used. ICS use could not be seen to significantly affect the lung mycobiome. These findings should be confirmed in other study populations before we can conclude that ICS use has no harmful effect on the lung mycobiome.

## Supporting information

**S1 Fig.** Percentage of reads belonging to Ascomycota/Basidiomycota in (A) oral wash and (B) bronchoalveolar lavage.
(PDF)

**S2 Fig.** Yue-Clayton measures from (A) controls and (B) participants with COPD. YC: Yue-Clayton measure. OW: oral wash, NCS: negative control sample, BAL: bronchoalveolar lavage. A Yue-Clayton measure of 0 means identical sample pairs, while a Yue-Clayton measure of 1

means unidentical sample pairs.
(PDF)

**S3 Fig.** Principal coordinates analysis plots by (A) study group and (B) inhaled steroids use. Differences in beta diversity were tested with permuted analysis of variance (PERMANOVA) adjusted for sex, age, and percentage of predicted FEV1 (permutations = 10000). No significant differences were seen in spread/dispersion (permutations = 1000).
(PDF)

**S4 Fig.** Principal coordinates analysis plots by sample type (A) before and (B) after symmetric Procrustes transformation. OW: oral wash, BAL: bronchoalveolar lavage. Arrows are drawn from the OW sample to the BAL sample from the same participant. Non-randomness ("significance") between the two configurations was tested with the protest function including the three first axis from the PCoA and specifying 999 permutations.
(PDF)

**S5 Fig. Plot of Qubit concentrations and comparisons between sample types.** BAL: bronchoalveolar lavage, NCS: negative control sample, OW: oral wash. Statistical differences in Qubit concentrations were tested using Wilcoxon signed-rank test as a paired test.
(PDF)

**S6 Fig. Alpha diversity plots and comparisons between sequencing runs.** BAL: bronchoalveolar lavage, OW: oral wash. Alpha diversity was evaluated using Shannon index. Statistical differences in alpha diversity were tested using Kruskal-Wallis.
(PDF)

**S7 Fig. Principal coordinates analysis plots divided by sequencing run.** OW: oral wash, BAL: bronchoalveolar lavage.
(PDF)

**S1 Table. Presumed fungal contaminants identified by Decontam in the MicroCOPD study.** ASV: amplicon sequence variant. The R package "Decontam" identified the ASV IDs above as contaminants. ASVs presumed to be contaminants were removed prior to analyses.
(PDF)

**S2 Table. Differential abundance/distribution testing on fungi in the MicroCOPD study using ANCOM v2, MicrobiomeDDA, and ALDEx2.** ANCOM v2: the second version of analysis of composition of microbiomes, MicrobiomeDDA: Microbiome Differential Distribution Analysis omnibus test, ALDEx2: the second version of ANOVA-Like Differential Expression, OW: oral wash, BAL: bronchoalveolar lavage. The most conservative value in ANCOM v2 has been used in the analyses (i.e. 0.9). Significance level = 0.05. Never- and ex-smokers were merged into non-smokers. The ALDEx2 approach works poorly if there are only a small number of taxa (less than about 50), so some groups were not analysed.
(PDF)

**S3 Table. Taxonomy and diversity comparisons of selected clinical variables in the Micro-COPD study divided by sample type and study group.** PERMANOVA: permuted analysis of variance, OW: oral wash, BAL: bronchoalveolar lavage, AN: ANCOM v2, M: MicrobiomeDDA, AL: ALDEx2, sign: significant, FEV$_1$: forced expiratory volume in 1 second. Analyses on FEV$_1$ were omitted for each study group separately due to a majority of controls having above 80% of predicted, and a majority of participants with COPD having below 80% of predicted. Diversity analyses on smoking habits in BAL samples from controls were omitted due to a lack of current smokers. Analyses on smoking habits were done by comparing current vs

non-current smokers.
(PDF)

**S4 Table. Summary of read/sequence counts in the MicroCOPD study.** NCS: Negative control sample, OW: oral wash, BAL: bronchoalveolar lavage, DADA2: Divisive Amplicon Denoising Algorithm version 2.
(PDF)

**S5 Table. Beta diversity comparisons using Bray-Curtis and Jaccard distances.** Comparisons were done (A) merged and (B) pairwise. OW: oral wash, BAL: bronchoalveolar lavage, yrs: years. Differences in beta diversity were tested with permuted analysis of variance (PERMANOVA) adjusted for sample type, study group, sex, and age (permutations = 10000).
(PDF)

**S6 Table. Permuted multivariate analysis of beta-dispersion using Bray-Curtis and Jaccard distances.** Comparisons were done (A) merged and (B) pairwise. OW: oral wash, BAL: bronchoalveolar lavage.
(PDF)

**S7 Table. Differential abundance/distribution testing on sequencing run using ANCOM v2, MicrobiomeDDA, and ALDEx2.** ANCOM v2: the second version of analysis of composition of microbiomes, MicrobiomeDDA: Microbiome Differential Distribution Analysis omnibus test, ALDEx2: the second version of ANOVA-Like Differential Expression, OW: oral wash, BAL: bronchoalveolar lavage. The ALDEx2 approach works poorly if there are only a small number of features (less than about 50). The most conservative value in ANCOM v2 has been used in the analyses (i.e. 0.9).
(PDF)

**S1 File. Bioinformatic processing.**
(PDF)

## Acknowledgments

The MicroCOPD is a large study with many co-workers. The authors wish to give their thanks to Per Sigvald Bakke, Harald G Wiker, Øistein Svanes, Sverre Lehmann, Marit Aardal, Tuyen Hoang, Tharmini Kalananthan, Randi Sandvik, Eli Nordeide, Hildegunn Bakke Fleten, Tove Folkestad, Ane Aamli Gagnat, Solveig Tangedal, Kristina Apalseth, Stine Lillebø, and Lise Østgård Monsen (Haukeland University Hospital and University of Bergen).

## Author Contributions

**Conceptualization:** Einar M. H. Martinsen, Tomas M. L. Eagan, Rune Nielsen.

**Data curation:** Einar M. H. Martinsen, Tomas M. L. Eagan, Elise O. Leiten, Rune Nielsen.

**Formal analysis:** Einar M. H. Martinsen, Andreu Paytuví-Gallart.

**Funding acquisition:** Tomas M. L. Eagan, Rune Nielsen.

**Investigation:** Einar M. H. Martinsen, Tomas M. L. Eagan, Elise O. Leiten, Ingvild Haaland, Gunnar R. Husebø, Kristel S. Knudsen, Christine Drengenes, Walter Sanseverino, Andreu Paytuví-Gallart, Rune Nielsen.

**Methodology:** Einar M. H. Martinsen, Tomas M. L. Eagan, Rune Nielsen.

**Project administration:** Tomas M. L. Eagan, Rune Nielsen.

**Software:** Einar M. H. Martinsen, Andreu Paytuví-Gallart.

**Supervision:** Tomas M. L. Eagan, Rune Nielsen.

**Writing – original draft:** Einar M. H. Martinsen.

**Writing – review & editing:** Einar M. H. Martinsen, Tomas M. L. Eagan, Elise O. Leiten, Ingvild Haaland, Gunnar R. Husebø, Kristel S. Knudsen, Christine Drengenes, Walter Sanseverino, Andreu Paytuví-Gallart, Rune Nielsen.

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
