## [Decision Letter · Decision Letter 0]

30 Dec 2020

PONE-D-20-36443

The pulmonary mycobiome - a study of subjects with and without chronic obstructive pulmonary disease

PLOS ONE

Dear Dr. Martinsen,

Thank you for submitting your manuscript to PLOS ONE. After careful consideration, we feel that it has merit but does not fully meet PLOS ONE’s publication criteria as it currently stands. Therefore, we invite you to submit a revised version of the manuscript that addresses the points raised during the review process.

Please note that there are significant issues raised during the review process focused on the technical aspects of the data acquisition including controls that need to be addressed (please see reviewers comments below). Data availability has to also comply with PLOS policies and our reviewers were not able to access your raw data. Please ensure all of this and he below comments are fully addressed in any revision.

We look forward to receiving your revised manuscript.

Kind regards,

Sanjay Haresh Chotirmall, MD PhD

Academic Editor

PLOS ONE

2. We noted in your submission details that a portion of your manuscript may have been presented or published elsewhere. ("Parts of the content of this manuscript has been presented at the National Microbiota Conference in Norway, and an abstract was published in an online programme. It was also accepted for an oral presentation at the ERS International Congress in 2019, and published as an abstract in the European Respiratory Journal.") Please clarify whether this conference was peer-reviewed and formally published. If this work was previously peer-reviewed and published, in the cover letter please provide the reason that this work does not constitute dual publication and should be included in the current manuscript.

"I have read the journal's policy and the authors of this manuscript have the following competing interests:

EMHM, EOL, IH, GRH, KSK, CD, WS, and APG declare no conflict of interest.

RN reports grants from The endowment of timber merchant A. Delphin and wife (The Norwegian Medical Association) and grants from GlaxoSmithKline during the conduct of the study, and grants and personal fees from AstraZeneca, grants and personal fees from GlaxoSmithKline, grants and personal fees from Boehringer Ingelheim, and grants from Novartis outside the submitted work.

TMLE reports grants from Helse Vest (Western Norway Regional Health Authority) during the conduct of the study, and personal fees from Boehringer Ingelheim outside the submitted work."

4. Thank you for stating the following in the Financial Disclosure section:

"The MicroCOPD study was funded by unrestricted grants and fellowships from Helse Vest, GlaxoSmithKline, Bergen Medical Research Foundation, and the Endowment of Timber Merchant A. Delphin and Wife through the Norwegian Medical Association.

We note that one or more of the authors are employed by a commercial company: 3Sequentia Biotech SL, Carrer D'Àlaba 61, 08005, Barcelona, Spain

(1)Please provide an amended Funding Statement declaring this commercial affiliation, as well as a statement regarding the Role of Funders in your study. If the funding organization did not play a role in the study design, data collection and analysis, decision to publish, or preparation of the manuscript and only provided financial support in the form of authors' salaries and/or research materials, please review your statements relating to the author contributions, and ensure you have specifically and accurately indicated the role(s) that these authors had in your study. You can update author roles in the Author Contributions section of the online submission form.

(2) Please also provide an updated Competing Interests Statement declaring this commercial affiliation along with any other relevant declarations relating to employment, consultancy, patents, products in development, or marketed products, etc.  

Reviewers' comments:

Reviewer's Responses to Questions

**Comments to the Author**

1. Is the manuscript technically sound, and do the data support the conclusions?

Reviewer #1: Yes

Reviewer #2: Partly

2. Has the statistical analysis been performed appropriately and rigorously? 

Reviewer #1: Yes

Reviewer #2: Yes

3. Have the authors made all data underlying the findings in their manuscript fully available?

Reviewer #1: No

Reviewer #2: No

4. Is the manuscript presented in an intelligible fashion and written in standard English?

Reviewer #1: No

Reviewer #2: Yes

5. Review Comments to the Author

Reviewer #1: Martinsen EMH et al; evaluated the oral wash and BAL mycobiome in COPD (n=93) and control subjects (n=100) from a single center in Western Norway. Candida was the dominant genus in all samples with higher abundance in oral compared to BAL samples. No difference in alpha and beta-diversity between study groups and ICS used.

This is the first study using BAL to access the airway mycobiome in COPD, however there were numerous weakness which need to be address

Major comments

• The control subjects were defined as no COPD and other lung diseases, however 22 of the control subjects had FEV/FVC <0.7, is this group of patient smoker? why are their lung function obstructive? Any underlying bronchiectasis or asthma which explained the abnormal lung function? As their spirometry is abnormal is hard to believe that these individuals have no underlying lung diseases, perhaps they should be removed from the control group and reanalyzed.

• The author mention that PBS was obtained as negative control samples, and contamination removed with Decontam, but it seems that negative control samples are also dominated by Candida (Figure S2), what was the abundance of Candida in the negative control samples?

• What is the clinical relevance and impact of this study finding?

• Was the sample collected at stable COPD? how many weeks post exacerbations were patients allow to be recruited for the study?

• Is there any patient on long term oral corticosteroid, antifungal or antibiotics?

• What is the breakdown of patient in each group (Figure 4) how many patients in COPD grade I/II and COPD grade III/IV?

• Please ensure sequencing data is publicly available

• The weakness including cross-sectional nature, interaction between various kingdom and host responses were not access in this study and should be discuss

Reviewer #2: General comments:

The authors present an ITS amplicon sequencing study of involving COPD patients and controls (n = 93 vs n = 100). From each subject, mycobiome profiles were derived for BAL, oral was, and a negative controls. While, the analysis of this data reveals difference between the BAL and Oral rinse samples, control and COPD patients do not significantly differ. No association between the mycobiome and clinical outcome is observed.

I have some specific comments and suggestions below;

1. What DNA yields were achieved in samples (OW and BAL) vs negative extraction blanks? This is a critical point. This data should be plotted and included in the manuscript. Both the DNA and fungal yields need to be quantified in some way to demonstrate that the sampling technique actually works i.e. a significantly higher DNA/fungal/Fungal DNA yield is obtained in samples relative to background contamination. Any samples where this cannot be safely concluded should probably be excluded.

2. What was the total raw sequencing depth and total read count in samples (OW and BAL) vs negative extraction blanks (NCS)? Please include a figure/table. Figure S2 shows only relative abundance i.e. the stacked bar charts are proportionally scaled and there is no way to determine how many reads were detected in the NCS compared to samples. A single figure representing aggregate read numbers for each taxonomic classification (genus level) for OW NCS and BAL would be useful. The multiple graphs in figure S2 are informative but a little difficult to digest at first glance. An accompanying graph summarising the aggregate comparison would help – i.e. Total read counts for NCS, OW and BAL for all reported OTUs.

3. For many subjects, negative controls (NCS) are highly similar to the samples (BAL/OW). Unless there is some other evidence for a signal in these samples (i.e. much higher DNA yield/read count in the samples vs NCS) then these must be excluded from analysis. If NCS profiles match either a BAL or OW sample and have comparable yield in terms of DNA yield or total number of classified reads then there is no way to tell whether this signal is coming from the sample or represents stochastic background contamination signal. Such cases should be systematically identified, excluded and the analysis repeated. This could impact interpretation regarding clinical association.

4. Data and code repository not accessible (https://doi.org/10.5061/dryad.w3r2280nz)

5. Batch effects. The authors assessed batch effects associated with sequencing runs. Assuming the DNA extractions were not all done as a single batch, the same analysis should be performed for DNA extraction batch effects (between-batch effects).

6. Confusing statement in the discussion concerning “medians were dominated by zeros, comparable to the 0.03 found in the CF study”. Please revise and state clearly what is meant by this without requiring the reader to refer to another paper.

7. Final paragraph; “still in its infancy” > “still in their infancy”

6. PLOS authors have the option to publish the peer review history of their article (what does this mean?). If published, this will include your full peer review and any attached files.

Reviewer #1: No

Reviewer #2: No

---

## [Author Response · Author response to Decision Letter 0]

12 Feb 2021

Thank you for the valuable comments to our paper, and for giving us the possibility to submit a revised version. We do believe that we have been able to address all comments from the reviewers, and that these comments have led to improvement of the manuscript. I have not submitted a new cover letter, as this was not specifically mentioned in the Decision Letter, but included the initial cover letter as this was mandatory by Editorial Manager to proceed with the submission. All information including updated COI and Funding statements, for both editor and reviewers, is found in the Response to Reviewers document which could be considered an updated cover letter. Please note that the temporarily URL for the review process is:

https://datadryad.org/stash/share/1V9eIhBEDmwNdcOqTGfyewO_lcfQlhlyodHDiizTC0U

I have had some issue by clicking the link above in the PDF built by Editorial Manager, but the link works fine if you copy and paste into a Firefox Internet browser or follow the link at the end of the PDF.

Regarding the content permission form requested upon revision as of 12th of February 2021:

I contacted the European Respiratory Journal and asked them to complete the form to which they replied:

For ERS congress abstracts, authors retain copyright. This is stated in the footnote of the published abstract.

You therefore do not need to request permission from ERS.

We hope that you find the revision satisfactory.

---

## [Decision Letter · Decision Letter 1]

9 Mar 2021

The pulmonary mycobiome - a study of subjects with and without chronic obstructive pulmonary disease

PONE-D-20-36443R1

Dear Dr. Martinsen,

We’re pleased to inform you that your manuscript has been judged scientifically suitable for publication and will be formally accepted for publication once it meets all outstanding technical requirements.

Kind regards,

Sanjay Haresh Chotirmall, MD PhD

Academic Editor

PLOS ONE

Additional Editor Comments (optional):

Reviewers' comments:

Reviewer's Responses to Questions

**Comments to the Author**

1. If the authors have adequately addressed your comments raised in a previous round of review and you feel that this manuscript is now acceptable for publication, you may indicate that here to bypass the “Comments to the Author” section, enter your conflict of interest statement in the “Confidential to Editor” section, and submit your "Accept" recommendation.

Reviewer #2: All comments have been addressed

2. Is the manuscript technically sound, and do the data support the conclusions?

Reviewer #2: Yes

3. Has the statistical analysis been performed appropriately and rigorously? 

Reviewer #2: Yes

4. Have the authors made all data underlying the findings in their manuscript fully available?

Reviewer #2: Yes

5. Is the manuscript presented in an intelligible fashion and written in standard English?

Reviewer #2: Yes

6. Review Comments to the Author

Reviewer #2: The authors have provided a detailed and extensive rebuttal that has satisfactorily addressed my concerns in so far as possible. I comment the authors on the thoroughness of the response.

7. PLOS authors have the option to publish the peer review history of their article (what does this mean?). If published, this will include your full peer review and any attached files.

Reviewer #2: No

---

## [Editor Report · Acceptance letter]

12 Mar 2021

PONE-D-20-36443R1 

The pulmonary mycobiome - a study of subjects with and without chronic obstructive pulmonary disease 

Dear Dr. Martinsen:

I'm pleased to inform you that your manuscript has been deemed suitable for publication in PLOS ONE. Congratulations! Your manuscript is now with our production department. 

Kind regards, 

on behalf of

Assistant Professor Sanjay Haresh Chotirmall 

Academic Editor

PLOS ONE